# Removal of Copper from Mining Wastewater Using Natural Raw Material—Comparative Study between the Synthetic and Natural Wastewater Samples

**Sonja Milićević** [1,*], **Milica Vlahović** [2], **Milan Kragović** [3], **Sanja Martinović** [2], **Vladan Milošević** [1], **Ivana Jovanović** [4] and **Marija Stojmenović** [3]

1   Institute for Technology of Nuclear and Other Mineral Raw Materials, 11000 Belgrade, Serbia; v.milosevic@itnms.ac.rs
2   Institute of Chemistry, Technology and Metallurgy, University of Belgrade, 11000 Belgrade, Serbia; mvlahovic@tmf.bg.ac.rs (M.V.); s.martinovic@ihtm.bg.ac.rs (S.M.)
3   Department of Materials, "VINČA" Institute of Nuclear Sciences-National Institute of the Republic of Serbia, University of Belgrade, 11000 Belgrade, Serbia; m.kragovic@vinca.rs (M.K.); mpusevac@vinca.rs (M.S.)
4   Mining and Metallurgy Institute Bor, 19210 Bor, Serbia; ivana.jovanovic@irmbor.co.rs
*   Correspondence: s.milicevic@itnms.ac.rs; Tel.: +381-64-116-3317

**Abstract:** The intent in this paper is to define how the batch equilibrium results of copper removal from a synthetic solution on natural zeolite can be used for prediction of the breakthrough curves in the fixed-bed system for both a synthetic solution and wastewater. Natural zeolite from the Vranjska Banja deposit, Serbia, has been fully characterized (XRD, chemical composition, DTA/TG, SEM/EDS) as a clinoptilolite with cation exchange capacity of 146 meq/100 g. The maximum adsorption capacity ($q_m$) in the batch of the mono-component system (synthetic copper solution) obtained using the Langmuir isotherm model was 7.30 and 6.10 mg/g for particle size below 0.043 and 0.6–0.8 mm, respectively. Using the flow-through system with the 0.6–0.8 mm zeolite fixed-bed, almost double the adsorption capacity (11.2–12.2 mg/g) has been achieved in a saturation point for the copper removal from the synthetic solution, compared to the batch. Better results are attributed to the constant high concentration gradient in flow-through systems compared to the batch. The complex composition of wastewater and large amounts of earth alkaline metals disturb free adsorption sights on the zeolite surface. This results in a less effective adsorption in flow-through systems with adsorption capacity in breakthrough point of 5.84 mg/g ($\sim 0.95 \times qm$) and in a saturation point of 7.10 mg/g ($\sim 1.15 \times qm$).

**Keywords:** zeolite; wastewater; heavy metals; batch; fixed-bed; adsorption capacity

## 1. Introduction

Copper is one of the most present metals in the industry due to its widest application compared to all non-ferrous metals (electroplating, power and electronics industry, light industry, machine industry, architecture, petrochemical industry, etc.). The copper mine plants are designed with the capacity that is more than ten times higher compared with the other heavy metals facilities. Therefore, there is a growing interest in solving the environmental issues of the copper mine industry especially because the heavy metals are non-biodegradable and tend to accumulate in living organisms causing serious diseases and disorders [1]. According to the World Health Organization standards (WHO), the tolerable limit of Cu in drinking water is 2.0 ppm [2,3].

Mining wastewater is an undesirable side effect of the exploitation of copper ore. During the excavation of the deposit, self-leaching of copper oxide structures frequently occurs, both at the open-pit mine as well as during underground exploitation. Disturbance in the balance of the rock mass

after excavation plays a decisive role in the self-leaching phase. Subsequently, oxidation of copper sulfide minerals occurs on the surface of the rock mass. The change in the structure of the terrain due to the excavation of ore from the rock mass, often results in the emergence groundwater. This creates the conditions for the dissolution of copper oxide minerals in these parts surface of the deposit, producing "blue waters"—mining wastewater with increased copper content. The properties of these wastewaters directly depend on the characteristics of the part of the deposit that is being exploited and the conditions on the surface: Copper content in the ore, mode of occurrence and structure of copper minerals, accompanying useful components, degree of surface oxidation and atmospheric conditions.

In order to optimize the excavation process at the open-pit and in the underground pit, it is necessary to perform efficient drainage of the deposit whilst the technical operations in the field remain uninterrupted. In the process of deposit drainage, vast quantities of "blue water" are transported from the excavation site, which is especially pronounced during periods of heavy precipitation and melting snow.

Many investigations have been undertaken with the aim of removing heavy metals from the wastewater streams, making it fit for reuse or release back to the environment. Removal of such pollutants from water discharges potentially dangerous substances in concentrations that impose their further treatment or disposal. The paradoxical outcome is that regulations designed to reduce water pollution have resulted in the need to deal with more hazardous waste. Due to the high amount of mining wastewaters, their continuous generation and disposal, noticeable amounts of copper are lost. Valorization and utilization of copper from these waste streams is of great importance as production of metal from primary raw materials becomes more and more difficult [4].

In order to develop less expensive, more effective, and environmentally friendly technologies, several approaches have been taken in recent times. However, all proposed technologies, even if they provide sufficient quality of the effluent, classify the copper contaminated water as a waste. Surely, copper released into the environment should not be observed only as the environmental ballast but also as the loss of the valuable non-renewable raw material. This is especially relevant in a case of the copper mine industry which produces many wastewater streams that contain the high concentration of copper. Due to the high copper price on the stock market and constantly growing demand for this metal on the market, minimizing the losses of copper becomes a very important issue. Therefore, many studies are occupied with developing the way that can provide both, removing the copper from wastewater and its valorization. Among common purification techniques, adsorption has gained much recent attention in this field because it is an eco-friendly, cost-effective, and straightforward operating technology.

In recent years, much attention has been focused on the selection and/or production of low-cost adsorbents with good metal-binding capacities [5–8]. Many investigations in this field pointed out the zeolite as a cost-effective adsorbent because of its good metal binding capacity, local availability in large quantities, an easy form of technology to operate, and avoidance of secondary pollution.

Zeolites are adsorbents derived from the vitroclastic sediments that have been altered diagenetically in subaerial and marine environments [9]. There are more than 40 naturally occurring zeolite frameworks and clinoptilolite is among the most abundant one. The primary building units of the zeolites are the $SiO_4$ tetrahedra that are mutually linked over the common oxygen atoms making the framework structure rich in cages, cavities, and channels. However, sometimes the $Si^{4+}$ is substituted by other metal ions and in a large number of cases that ion is an $Al^{3+}$. This results in a negative net charge of the zeolite framework. This negative charge is compensated by positive extra-framework ions which are located in the so-called charge-compensating positions, where they neutralize the negative charge. The charge compensation is usually provided by the ions such as $Na^+$, $K^+$, $Ca^{2+}$, $Mg^{2+}$ [10,11], and the presence of these ions gives zeolite the cation exchange capacity (CEC). Thanks to the favorable ion exchange behavior of the zeolite, this mineral can be regenerated in the form suitable to be reused. The by-product of the regeneration process is the highly concentrated copper solution that can be further treated in order to create value-added products from the recovered adsorbed metal.

In order to increase the zeolite adsorption capacity toward a specific contaminant, iron-based modification of the zeolite has been the most investigated [12–14]. Many attempts were made to synthesize mixed systems of Fe oxides and zeolites and those systems were proved to retain greater amounts of heavy metals compared to the parent material [15–17]. The formation of a Fe-oxide phase on the external surface of the zeolite crystallites increases the number of active sites on the adsorbent because of the presence of the Fe-OH groups [18]. Unlike unmodified zeolites, where the main mechanism responsible for heavy metal removal is the cation exchange, in Fe-zeolite composites the mechanism is complex and plays both a cation exchange and binding for active centers of Fe-oxide phase. Consequently, the regeneration of modified zeolites is still uninvestigated and present main obstacles for their sustainable application in wastewater treatments.

One of the world's most abundant copper ore is the chalcopyrite-copper iron sulfide mineral and, therefore, Fe ions are the most presented accompanying ions in mining waste. While the presence of Fe ions is not desirable from the adsorption efficiency point of view, the Fe-oxide particles, as previously mentioned, are the ones that have a high affinity toward heavy metals. However, this procedure requires an extremely high pH value [12,15,18], meaning that Fe precipitation in the mild conditions, that will remain copper in a solution, would not result in this supplementary effect. Some further investigations should be conducted in a direction toward producing Fe-oxide particles under mild conditions. This would result in removing Fe ions from wastewater and increasing the zeolite capacity for heavy metals by coating the zeolite with Fe-oxide particles.

The obtained results with natural and modified zeolites provided numerous findings in the field of kinetic study, phenomena that are responsible for the removal, influence of the different conditions (pH, initial concentration, particle size), etc. However, all these data are obtained using synthetic solutions containing only copper.

This paper presents details of investigations that have been developed to point out the differences between the results obtained in pure synthetic solutions and in natural complex water streams. Unlike a synthetic solution (mono or binary system), the composition of the wastewater is complex. A wide range of heavy metals, alkali and alkaline earth cations, as well as different anions are present in mining wastewater. For that reason, learnings from the removal of the copper from a synthetic solution, may not necessarily apply in a complex system such as mining wastewater.

The focus in this paper will be on the transfer of acquired knowledge from the laboratory investigations on the natural industrial wastewater. Especially, how to reduce the negative influence of the present accessory ions on the copper removal and to decrease the differences in the purification level between the synthetic solutions and the industrial wastewater will be investigated. There is a lack of scientific data on the removal of heavy metal ions from wastewater using zeolites. These investigations can form the foundations for further research and scientific learnings that will underpin industry applications.

## 2. Materials and Methods

### 2.1. Material Characterization

#### 2.1.1. Clinoptilolite

Raw zeolite, clinoptilolite (Cli) from the Zlatokop deposit (Vranjska banja) in Serbia was used as an absorbent. After crushing and grinding, the sample was sieved to the fractions:

- Below 0.043 mm—optimal for the zeolite characterization and investigations of maximum adsorption capacity.
- 0.6–0.8 mm—optimal for the flow-through experiments due to better hydraulic properties than smaller (micron) fractions.

The applied principal methods of zeolite identification and characterization were X-ray powder diffraction (XRPD), scanning electron microscopy (SEM), differential thermal analysis (DTA), chemical analysis, and total cation exchange capacity (CEC).

A scanning electron microscope (SEM) equipped with energy dispersive X-ray spectroscopy (EDS) (JSM-6610LV, JEOL, Tokyo, Japan) was used for microstructure analysis of the sample.

Thermal analyses were conducted on the samples previously kept in a relative atmospheric humidity of 75% for 24 h. The thermogravimetric/differential thermal (TG/DT) analysis (STA-409 EP, Netzsch, Selb, Germany) was carried out in an air atmosphere in a temperature range of 20–1000 °C. The heating rate was 10 °C/min. DTA results, as previously reported, showed a typical curve for clinoptilolite with endothermic peak around 125 °C referred to the process of dehydration [19].

The chemical composition of natural clinoptilolite was determined by atomic absorption spectrophotometry (AAS, PinAAcle 900T, Perkin Elmer, Waltham, MA, USA).

The total cation exchange capacity of the zeolitic tuff was determined using the standard procedure [20] with 1 M $NH_4Cl$ (Sigma-Aldrich, Munich, Germany). The amount of released cations was measured by atomic absorption spectrophotometry using the same AAS device.

### 2.1.2. Wastewater

Synthetic solutions were prepared using $CuSO_4 \cdot 5H_2O$ p.a. (Lach:ner, Neratovice, Czech Republic).

Industrial wastewater originates from the Serbia Zijin Copper d.o.o. Bor (former RTB Bor). This mining and smelting complex annually produces over 35,000 tons of copper. Figure 1 presents the Cerovo Lake from which water samples were taken for these experiments.

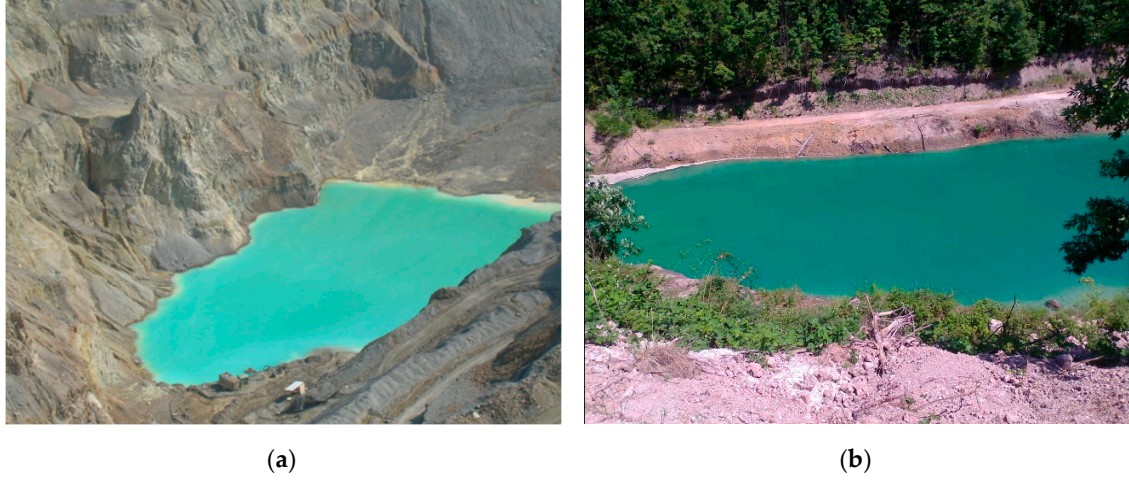

(**a**)                                                    (**b**)

**Figure 1.** ZIJIN Bor "blue water". (**a**) Cerovo deposit open pit and (**b**) Cerovo accumulation lake, where the wastewater has been transported.

The main problem with the sample is the very low pH value that greatly influences the adsorption capacity due to the high concentration of $H^+$ ions, which is competitive with the copper ion [21,22]. Therefore, the pre-treatment is needed in order to adjust the pH value to the one that less interferes with the copper adsorption. The pH correction of the wastewater has been performed using 1 and 0.1 M KOH p.a. (Sigma-Aldrich, St. Louis, MO, USA). Alkali was added to the wastewater dropwise until the pH value was 4.5. This pH is optimal for reduction of the $H^+$ ions and precipitation of the competitive Fe ions, both, with a high affinity towards zeolite, while copper and accompanying metals should remain in the solution.

The chemical composition of industrial wastewater and synthetic solution was determined by atomic absorption spectrophotometry using the same AAS device.

## 2.2. Adsorption Experiments

### 2.2.1. Batch System

Uptake of Cu from the synthetic solution in the batch system was investigated, for both zeolite fractions (below 0.043 and 0.6–0.8 mm) by shaking 1 g of Cli, with 50 mL of aqueous solution of $CuSO_4 \cdot 5H_2O$ p.a. (Lach:ner, Neratovice, Czech Republic) with various initial Cu concentrations (50–800 mg/L of $Cu^{2+}$). During the experiment, the pH value was in the range 4.5–5.0. Experiments were performed at room temperature using the batch technique for 24 h. After equilibration all suspensions were centrifuged and the concentrations of the remaining Cu in supernatants were determined using AAS.

Uptake of Cu from mine wastewater using the batch technique was performed by shaking 1 g of Cli with 50 mL of wastewater.

Knowing the exact liquid phase concentrations at the beginning of the process ($C_0$, mg/L) and after 24 h ($C_t$, mg/L) the equilibrium concentration of the sorbate in the solid phase ($q_e$, mg/g) was calculated:

$$q_e = \frac{V \times C_e}{m} = \frac{V \times (C_0 - C_t)}{m} \tag{1}$$

where $V$ (L) and $m$ (g) represent the volume of liquid phase and sorbent mass used in the reaction.

Usually, the equilibrium relationships between an adsorbent and an adsorbate are described by sorption isotherms, which represent the ratio between the amount adsorbed and that remaining in the solution at a fixed temperature under equilibrium conditions [23,24].

Experimental sorption data were simulated by Langmuir and Freundlich isotherm models, given by Equations (2) and (3), respectively [25–27]:

$$q_e = \frac{K_L\, C_e\, q_m}{1 + K_L\, C_e} \tag{2}$$

$$q_e = K_F\, C_e^{\beta} \tag{3}$$

where $q_e$ (mg/g) is the adsorbed amount per gram of adsorbent, $q_m$ (mg/g) is the monolayer maximum adsorption per gram of adsorbent, $C_e$ (mg/L) is the equilibrium concentration of pollutants in solution, $\beta$ is a correction factor implying heterogeneous surface, and $K_L$ (L/mg) and $K_F$ (L/mg) are Langmuir and Freundlich equilibrium constants of relative models.

The essential characteristics of the Langmuir isotherm can be expressed by a dimensionless constant called the separation factor $R_L$ [1,28]:

$$R_L = \frac{1}{1 + K_L C_0} \tag{4}$$

where $C_0$ is the initial concentration of the adsorbate (mg/g).

$R_L$ is related to the nature of adsorbent/adsorbate interaction, it designates the shape of the isotherm. Its values indicate that the adsorption is unfavorable when $R_L > 1$, linear when $R_L = 1$, favorable when $0 < R_L < 1$, and irreversible when $R_L = 0$.

Based on the knowledge of the Langmuir constant $K_L$, it is also possible to calculate the change in the Gibbs free energy ($\Delta G$) of the sorption process [29,30]:

$$\Delta G = -RT \ln K_L \tag{5}$$

where $R$ is the gas constant (8.314 J/mol·K) and $T$ [K] is the absolute temperature.

Gibbs free energy can be used to determine the spontaneity of a process. Reaction systems under normal conditions generally tend to reach a state of minimum free energy. Due to this natural tendency, a negative change in the Gibbs free energy ($\Delta G$) is a quantitative measure of the favourable

reaction potential. In other words, spontaneous reactions release energy. When $\Delta G < 0$, the process is spontaneous. When $\Delta G > 0$ the process is not spontaneous, instead, it will proceed spontaneously in the reverse direction.

### 2.2.2. Flow-through System

The experiments in flow-through (fixed-bed) systems, both, with synthetic and natural wastewater, were carried out in a glass column with the inner diameter of 1.2 cm. The column was filled with the 0.6–0.8 mm particle size zeolite sample. The zeolite bed height was 4.0 cm, corresponding to the bed volume of 45 cm$^3$. The fixed-bed was packed in a way leaving no air gaps between the zeolite particles. The experiments were carried out at isothermal conditions, with variations in initial concentrations of the synthetic solution and constant solution flow through the column ($Q = 1$ or 3 mL/min). The solution was continuously injected at the top of the column, passed through the zeolite fixed-bed, and the flow constancy was maintained using a vacuum pump. At selected time intervals, copper concentration was determined in the effluent. Moreover, copper adsorption from natural wastewater was realized in the same glass column, under the same conditions (flow, filter height, i.e., contact time). The process was stopped, when the Cu concentration in the effluent became equal to the initial concentration in the influent. After each adsorption cycle, the regeneration cycle was performed. The regeneration process was also performed in a fixed bed technique with 15 g/L solution of Na$_2$SO$_4$ (Lach:ner, Neratovice, Czech Republic). The passing of the solution in both, the adsorption and the regeneration cycle, was from the top to the bottom, where the effluent was collected and tested for the presence of the Cu$^{2+}$.

Obtained results were analyzed using breakthrough curves which are characterized by breakthrough and saturation points, and corresponding capacities. The capacity in the breakthrough point q$_b$ is defined as the amount of copper ions bound on zeolite when the concentration of copper in the effluent reaches 5% of the initial concentration [31]:

$$q_b = \frac{n_b}{m} = \frac{C_0 \times V_b}{m} \tag{6}$$

where $C_0$ is the influent concentration (mg/L), $V_b$ is the volume of solution passed in the breakthrough point, and m is the mass of the zeolite in fixed-bed.

The saturation point is defined as a moment when the concentration of the copper ions in the effluent reaches 95% of the initial value, and the capacity in the saturation point q$_s$ can be calculated [31]:

$$q_b = \frac{n_s}{m} = q_b + \frac{(C_0 - C_b) + (C_0 - C_s)}{2} \frac{(V_s - V_b)}{m} \approx q_b + \frac{0.95\,C_0}{2} \frac{(V_s - V_b)}{m} \tag{7}$$

where $C_0$ is the initial concentration (mg/L), $C_b$ is the concentration in the breakthrough point (mg/L), $C_S$ is the concentration in the saturation point, $V_s$ is the volume of solution passed in saturation point, $V$ is the volume of solution passed in the breakthrough point, and $m$ is the mass of the zeolite in fixed-bed.

## 3. Results and Discussion

### 3.1. Clinoptilolite Characterization

According to the semi quantitative X-ray powder diffraction (XRPD) analysis (Figure 2), the clinoptilolite content was around 75% among the crystalline phase. The accessory minerals were quartz (Q), feldspar (F), and carbonate (C), presented as impurities, but their effects on the physico-chemical behavior of zeolite were limited.

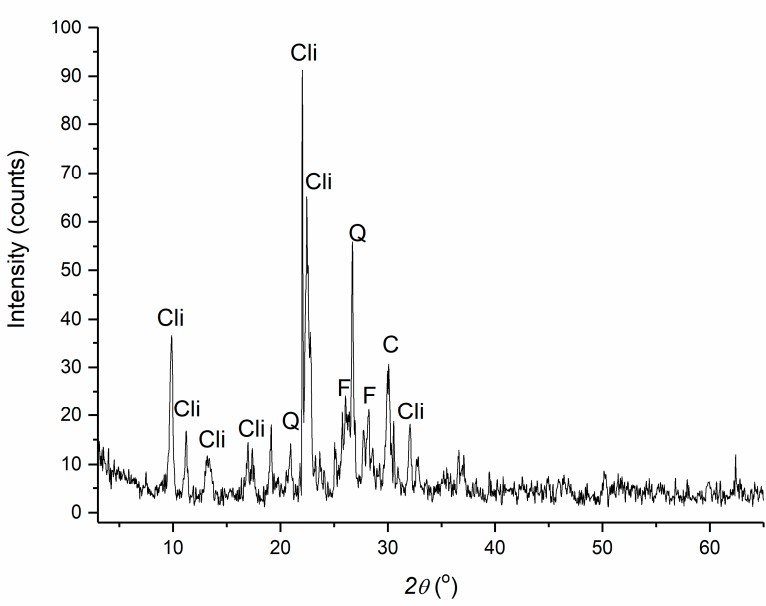

**Figure 2.** XRPD pattern for clinoptilolite (Cli—clinoptilolite, F—feldspar, Q—quartz, C—carbonate).

SEM images of used zeolite, taken with different magnifications, are presented in Figure 3. Crystal morphology with monoclinic symmetry typical for zeolite (clinoptilolite) can be seen in Figure 3a, while the presence of natural nanofibers is observed in Figure 3b due to higher magnification.

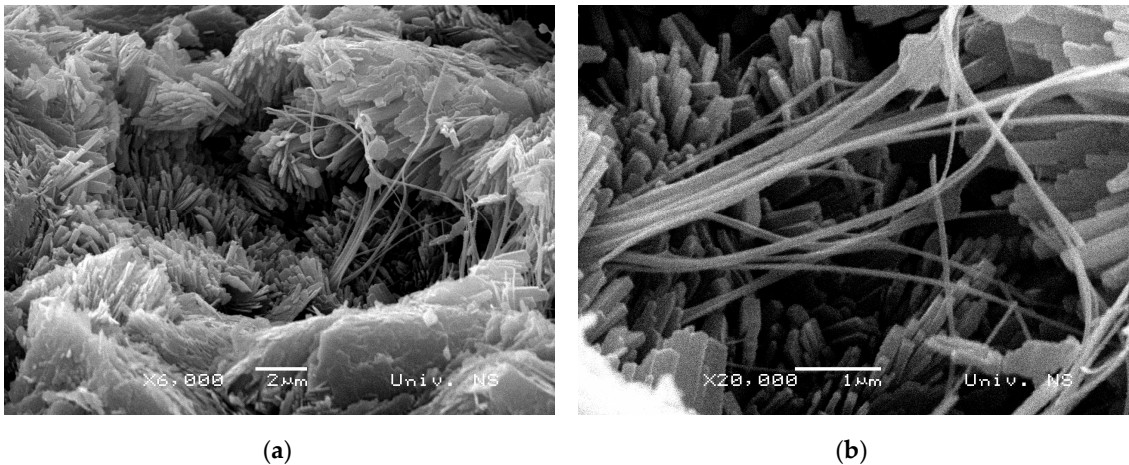

(**a**)                    (**b**)

**Figure 3.** SEM images of zeolite taken with different magnifications: (**a**) 6000×, (**b**) 20,000×.

The qualitative EDS analysis of clinoptilolite tuff and single clinoptilolite crystal is given in Figure 4, while Table 1 presents the chemical composition of clinoptilolite determined using AAS and EDS.

Based on the results shown in Figure 4 and Table 1, the high presence of Si and Al in clinoptilolite tuff, as well as the presence of Ca, K, Na, and Mg, typical for zeolite, are evident. Deviations in the values obtained using AAS and EDS are insignificant and can be attributed to the heterogeneity of the sample. Clinoptilolite is defined with the framework topology $5.5 > Si:Al \geq 4.0$ [32–34], and the investigated clinoptilolite is in accordance with literature data. According to the EDS analysis of a single clinoptilolite crystal, the Cli sample contained no structural Fe, so all of the Fe in the untreated Cli originated from the associated minerals [18].

The total cation exchange capacity of the zeolitic tuff was 146 meq/100 g, where calcium (85 meq/100 g) was the dominant exchangeable ion but Na (23.5 meq/100 g), Mg (22 meq/100 g), and K (15.5 meq/ 100 g) were also present in the starting zeolitic tuff as exchangeable cations.

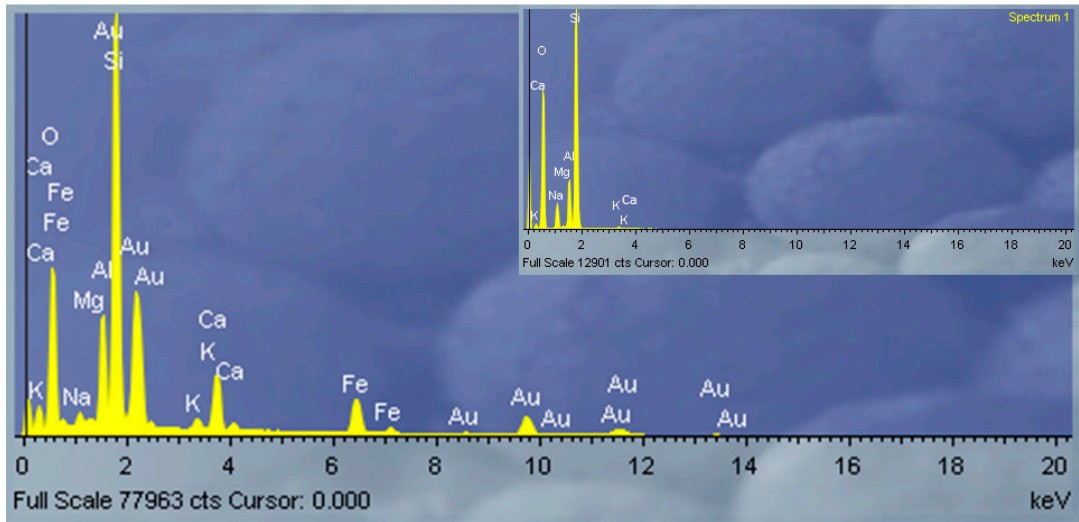

**Figure 4.** Qualitative EDS analysis of clinoptilolite tuff and single clinoptilolite crystal (insertation).

**Table 1.** Chemical composition of clinoptilolite tuff determined using atomic absorption spectrophotometry (AAS) and EDS.

|  | Content % | | | | | | | |
|---|---|---|---|---|---|---|---|---|
|  | **SiO$_2$** | **Al$_2$O$_3$** | **Fe$_2$O$_3$** | **CaO** | **MgO** | **K$_2$O** | **Na$_2$O** | **IOL** |
| AAS | 66.09 | 13.06 | 0.21 | 3.54 | 0.24 | 2.83 | 0.46 | 13.54 |
| EDS | 67.89 | 13.70 | 0.85 | 3.97 | 0.67 | 1.26 | 0.80 | 10.86 |

*3.2. Adsorption Experiments*

To study the copper adsorption from both synthetic and wastewater solutions by natural zeolitic tuff (Cli) in batch systems, the adsorption isotherm was determined.

3.2.1. Experiments with Synthetic Solutions

The isotherm for copper adsorption from synthetic solutions by Cli, obtained by plotting the equilibrium concentration of the remaining copper in the solution, against the amount of copper adsorbed per unit weight of adsorbent, is presented in Figure 5. Experiments were performed on Cli samples of two different particle sizes (below 0.043 and 0.6–0.8 mm). The influence of particle size plays an important role in adsorption experiments, since higher specific surface leads to higher adsorption. On the other side, in flow-through experiments due to low hydraulic conductivity of micro-particles, larger loading adsorbent size is preferable.

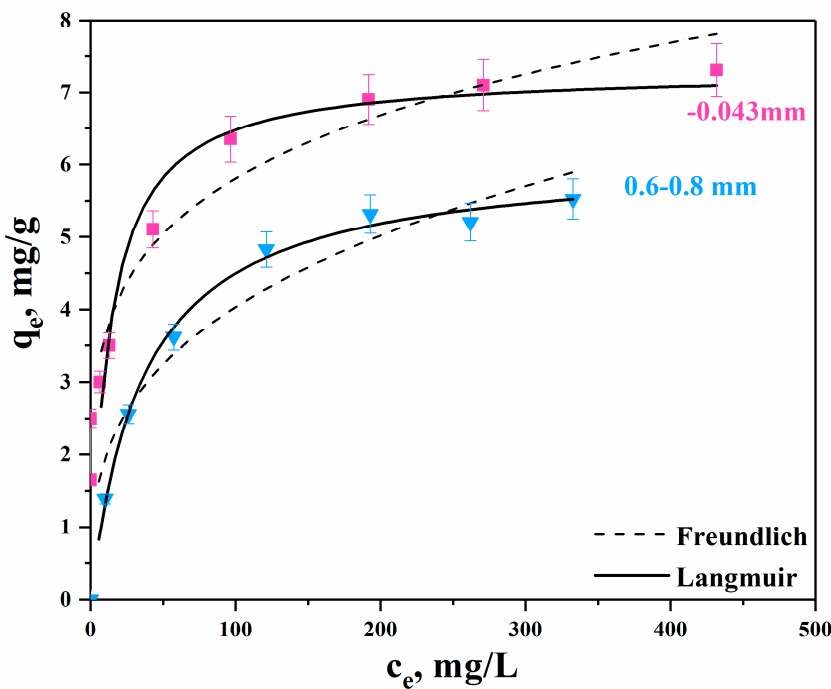

**Figure 5.** Adsorption isotherms of $Cu^{2+}$ from synthetic solutions on zeolite of different particle sizes, T = 298 K, $pH_0$ = 4.5.

Parameters and correlation coefficients of various isotherms for adsorption of $Cu^{2+}$ from synthetic solutions on zeolite of different particle sizes are presented in Table 2.

**Table 2.** Parameters and correlation coefficients of various isotherms for adsorption of $Cu^{2+}$ from synthetic solutions on zeolite of different particle sizes, T = 298 K.

| Freundlich | $R^2$ | $K_F$ (L/mg) $\times 10^{-3}$ | $\beta$ |
|---|---|---|---|
| −0.043 mm Cli | 0.951 | 1.31 | 0.203 |
| 0.6–0.8 mm Cli | 0.935 | 0.87 | 0.315 |
| **Langmuir** | $R^2$ | $q_m$ (mg/g) | $K_L$ (L/mg) |
| −0.043 mm Cli | 0.964 | 7.30 | 0.08 |
| 0.6–0.8 mm Cli | 0.994 | 6.10 | 0.03 |

As can be seen from Table 2, both models fit the experimental data with great precision. However, the Langmuir isotherm model fits the experimental data better than the Freundlich isotherm model and has a higher correlation coefficient ($R^2$) value. Hence, the adsorption of $Cu^{2+}$ onto Cli can be described as monolayer chemisorption on a homogeneous surface [28]. Based on the Langmuir model maximum adsorption capacity for particle size below 0.043 mm is 7.30 mg/g (0.12 mmol/g) and for 0.6–0.8 mm is 6.10 mg/g (0.10 mmol/g). The maximum adsorption capacity obtained from the Langmuir model is of great importance since it provides the data that simplify control of the flow-through experiments. These kinds of column experiments are time-lasting and approximate information about the expected moment when the breakthrough point occurs can reduce the manipulative time, efforts, and analysis.

For the extent of the investigated initial concentrations, the value of the separation factor $R_L$ is in the range of 0.01–0.17, which implies that the $Cu^{2+}$ adsorption onto zeolite is a favorable process. The values of the $R_L$ increase with the rise of the initial concentration, indicating that the favoring of the reaction also increases.

Based on the experimental results, the value of the Gibbs free energy is −21.1 kJ/mol, meaning that the copper adsorption onto zeolite is a spontaneous process.

In large wastewater treatment systems, the application of stationary (batch) procedures would require large volume contact reactors. Application of flow-through, i.e., column systems could greatly facilitate the process. The column procedure consists of alternating the operating and regeneration cycles of the filter. During the operating cycle, wastewater is passed through a layer of zeolite in the column until the concentration of the pollutant at the outlet is equal to the concentration at the inlet.

The experimental results for synthetic solutions in flow-through (column) systems are given and interpreted by a breakthrough curve, which is a plot of relative effluent concentration ($C/C_0$) against the duration of the experiment (*t*) (Figures 6 and 7).

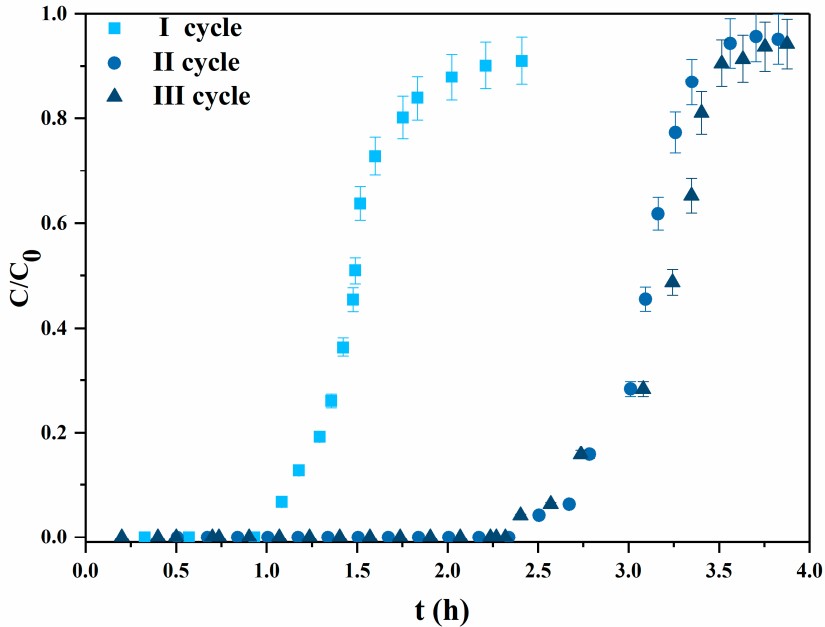

**Figure 6.** Breakthrough curves for the copper removal from the synthetic solution during I, II, and III operating cycles under the same conditions in a column, T = 298 K, pH$_0$ = 4.5.

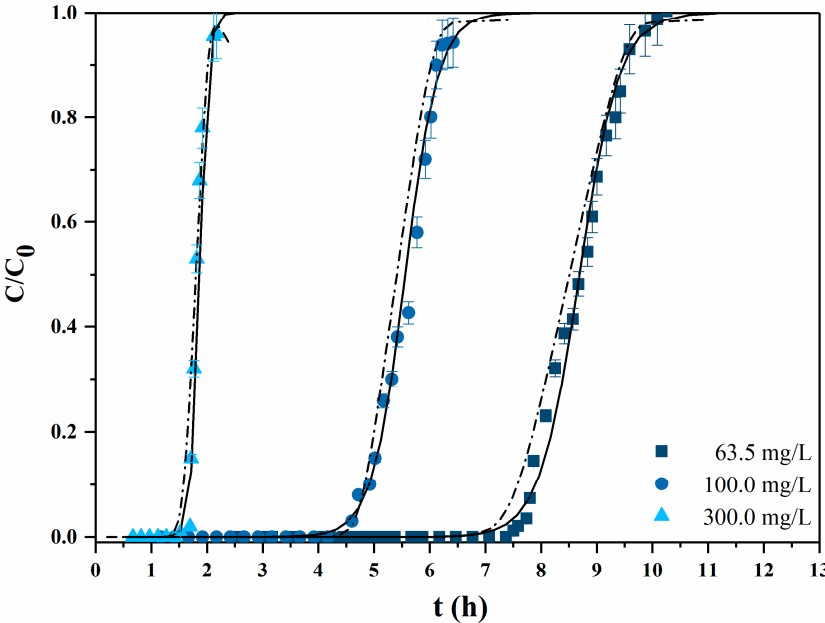

**Figure 7.** Breakthrough curves for the copper removal from the synthetic solution during IV, V, and VI operating cycles for the different initial concentrations, T= 298 K, pH$_0$ = 4.5.

Breakthrough curves for the copper removal from the synthetic solution during I, II, and III operating cycles under the same conditions in a column are presented in Figure 6.

Influence of the initial concentration on the adsorption capacity in breakthrough ($q_b$) and saturation ($q_S$) points was also investigated and the obtained results are presented in Figure 7.

Experimental conditions for six operating cycles and parameters calculated based on breakthrough curves from Figures 6 and 7 are presented in Table 3.

**Table 3.** Experimental conditions for operating cycles and values of parameters calculated from breakthrough curves.

|  | $c_0$ (mg/L) | $Q$ (ml/min) | $q_b$ (mg/g) | $q_S$ (mg/g) | $\eta$ (%) |
|---|---|---|---|---|---|
| I | 63.5 | 3 | 4.13 | 7.37 | 56.1 |
| II | 63.5 | 3 | 9.78 | 12.00 | 81.5 |
| III | 63.5 | 3 | 9.97 | 11.87 | 83.9 |
| IV | 63.5 | 1 | 9.91 | 11.24 | 88.1 |
| V | 100.0 | 1 | 9.72 | 11.62 | 83.6 |
| VI | 300.0 | 1 | 9.78 | 12.19 | 80.2 |

Based on the breakthrough curves under exactly the same initial conditions, given in Figure 6, and the data from Table 3, it is noticed that during the first operating cycle, the adsorption capacity of $Cu^{2+}$ ions is significantly lower than in the case of the next operating cycle. Namely, during the first operating cycle, the filter-bed is filled with natural zeolite, where cations such as $Na^+$, $K^+$, $Ca^{2+}$ and $Mg^{2+}$ are in exchangeable positions. After this cycle, saturated zeolite is regenerated with a concentrated $Na_2SO_4$ solution, after which the zeolite turns into mono-ionic Na-form. The type of cations present in clinoptilolite channels also determines the number of water molecules coordinated in the elemental cell. The monovalent cations, such as $Na^+$ bind fewer water molecules [35]. The reduced water content in the structure of clinoptilolite leads to less disturbances in the binding of $Cu^{2+}$ ions, compared to the structure in which divalent cations are also present. At the same time, during the first operating and recycling cycle the filter-bed is completely wetted, the zeolite particles are more compactly arranged, which additionally reduces the presence of cavities in the filter. This reduces the interspace between the particles, the fixed-bed becomes more compact and homogeneous, which results in a better and longer contact between the zeolite and the solution that needs to be purified [36]. The breakthrough curves of the operating cycles that follow the first one, are mainly overlapping at the key operating parameters such as $q_b$ and $q_S$ (Figure 6).

As seen in Figure 7, with the increase in the initial copper concentration the breakthrough and saturation points are moving to the left indicating the faster saturation of zeolite. Nevertheless, the total capacity in these two points remains almost the same, regardless of the initial concentration (Table 3). Slight differences in these values are not of great importance as the experimental curves are greatly overlapping with the theoretical one. This is of great importance for the application in semi-industrial and industrial conditions. The knowledge of working conditions of such system facilitates the managing of the column systems and enables control of the desired effluent.

If the maximum adsorption capacity from the batch experiment and the capacity in the saturation point ($q_S$) are compared, it can be concluded that the achieved adsorption capacity of ~11.80 mg/g (~0.19 mmol/g) in the flow-through experiments has almost two times higher value than the one in the batch systems that reached 6.10 mg/g (0.10 mmol/g). Concentration gradient between the adsorbate and adsorbents surface is a driving force for the adsorption. This means that a higher concentration gradient leads to a more efficient and faster adsorption. Unlike in discontinuous (batch), in continuous (flow-through) systems the solution constantly comes into contact with fresh adsorbent, i.e., zeolite where all cationic sites are free. Therefore, concentration gradient is always much higher in flow-through systems compared to the batch experiment, where the concentration gradient is high only at the initial point. Namely, in batch systems the concentration gradient decreases during the contact

time due to the occupation of the cationic sites and interferences. In fixed-bed systems, these effects are annulled.

### 3.2.2. Experiments with Mining Wastewater

The properties and behavior of mining wastewaters (known as "blue waters") greatly depend on the characteristics of the deposit that has been exploited (copper content and state, presence of associated minerals). Therefore, the quality of the "blue water" can greatly vary within the same deposit. Table 4 gives the results of the copper concentration and pH values of the "blue water" from ZIJIN Bor (former RTB Bor), taking account of the different parts of the deposit and the season conditions.

**Table 4.** Copper concentration and pH values from different parts of mine wastewater.

|               | 1    | 2    | 3    | 4    | 5      | 6    | 7    | 8    |
|---------------|------|------|------|------|--------|------|------|------|
| pH            | 1.89 | 2.43 | 2.87 | 4.75 | 3.65   | 3.89 | 3.68 | 4.22 |
| $[Cu^{2+}]$, mg/L | 198  | 180  | 70.8 | 44.1 | 1269.6 | 1928 | 963  | 338  |

Chemical composition of the initial wastewater sample from Cerovo deposit (labelled BW) and the one after the correction of pH value (labelled $BW_1$), as well as the results after adsorption using the batch technique (sample labelled Cli), are presented in Table 5.

**Table 5.** Chemical analysis of mining wastewater before pretreatment (BW), after pretreatment ($BW_1$), and after copper removal using the batch technique (Cli).

| Title  | Cu    | Fe   | Zn   | Mn   | Na   | K     | Ca    | Mg    | pH   |
|--------|-------|------|------|------|------|-------|-------|-------|------|
|        | (mg/L) |      |      |      |      |       |       |       |      |
| BW     | 205.0 | 14.6 | 27.6 | 45.1 | 84.7 | 1.27  | 480.0 | 442.0 | 2.89 |
| $BW_1$ | 201.0 | 1.21 | 25.3 | 43.1 | 85.2 | 91.5  | 476.0 | 445.0 | 4.5  |
| Cli    | 151.3 | /    | 19.8 | 40.9 | 90.3 | 108.3 | 490.0 | 456.0 | 4.7  |

From Table 5, it can be observed that the removal capacity of the copper from the $BW_1$ sample is 4.97 mg/g (0.08 mmol/g). Compared to the value of 7.30 mg/g (0.12 mmol/g) that is obtained with the synthetic solution under the same experimental conditions (amount of adsorbent, particle size, time, temperature) this removal capacity is remarkably lower. However, the presence of other heavy metals ions in the $BW_1$ sample must be taken into account, since they also compete for free adsorption sites. In order to compare values of the adsorbed heavy metals, their removal capacities must be expressed in milliequivalents (meq) or mmol, since they are all divalent. Based on this calculation, the adsorption capacity for Zn is 0.55 mg/g (0.01 mmol/g) and for Mn 0.22 mg/g (0.004 mmol/g), giving the total adsorption capacity for heavy metals of 0.094 mmol/g. Additional deviation from the value of 0.23 mmol/g, obtained in the mono-component system, should be attributed to the large presence of alkaline earth ions, such as $Ca^{2+}$ and $Mg^{2+}$. These ions also interfere with exchangeable cations and adsorption sites and all this should be taken into consideration before performing the experiments with natural wastewater samples. Finally, from the obtained results in batch systems, it can be concluded that the removal capacity of heavy metals from wastewater by natural zeolite is up to 80% of removal capacity from the monovalent systems.

Results from the flow-through experiments with $BW_1$ are presented in Figure 8. Experimental results are presented with dots, while the predicted breakthrough curve for the initial concentration of 200 mg/L (3.15 mmol/L) and 275 mg/L (4.33 mmol/L) are presented with the interrupted and full line, respectively. Namely, the interrupted predicted breakthrough curve is given for the solution that contains only copper with the initial concentration of 201 mg/L (3.16 mmol/L). The presence of $Zn^{2+}$ and $Mn^{2+}$ ions in concentrations of 25.3 mg/L (0.39 mmol/L) and 43.1 mg/L (0.78 mmol/L), respectively, also must be taken into account. When these concentrations get calculated to mmol and

then standardized to $Cu^{2+}$, the resulted concentration of the solution, standardized only to copper is 275 mg/g. According to the results in Figure 6, experimental data are more close to the predicted breakthrough curve for the standardized initial concentration of 275 mg/g, as expected. The deviation from this curve is associated with the presence of Ca and Mg, as previously stated.

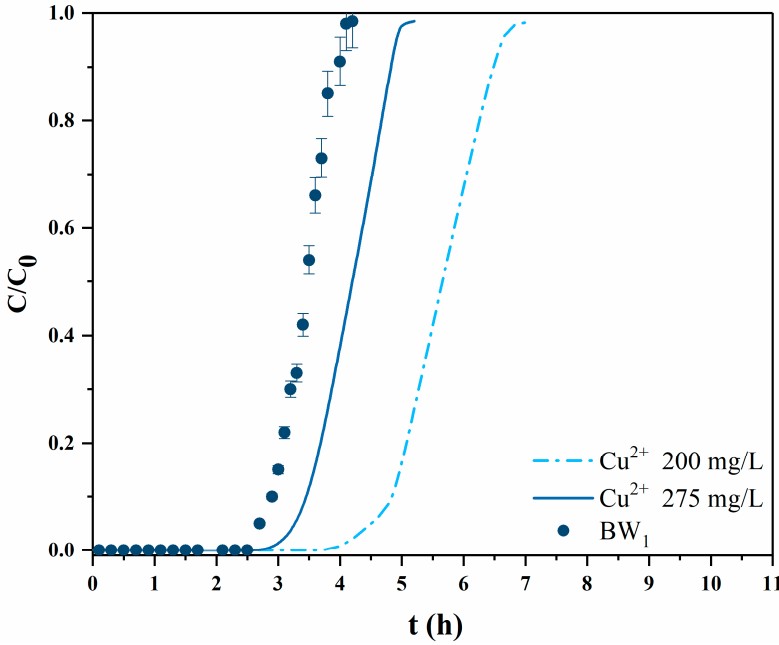

**Figure 8.** Breakthrough curves for the copper removal from the mine wastewater, T = 298 K, $pH_0$ = 4.5.

Experimental conditions and parameters calculated based on breakthrough curves from Figure 8 are presented in Table 6.

**Table 6.** Experimental conditions and values of parameters calculated from breakthrough curves.

| $c_0$ $(Cu^{2+})$ (mg/L) | Q $(cm^3/min)$ | $q_b$ (mg/g) | $q_s$ (mg/g) | η (%) |
|---|---|---|---|---|
| BW$_1$ 200.0 | 1 | 5.84 | 7.05 | 87.4 |

Upon the breakthrough curves for synthetic solution and wastewater, it can be concluded that the saturation point for heavy metals from wastewater by natural zeolite (7.05 mg/g) is up to 60% of saturation point from the monovalent systems (~11.8 mg/g). This indicates that the presence of accompanying ions disturb the removal of heavy metals in flow-through systems more than in the batch one. For a better understanding of this, it is necessary to involve the mass zone transfer MZT, which is defined as the zone of the packed column where the active adsorption happens. As the adsorption advances, MZT progresses down through the fixed-bed and when it comes to the bottom of the column, the breakthrough point occurs. For a mono-component system, this means that when a synthetic solution passes through MZT, after that zone almost an ion-free solution continues through the rest of the fixed-bed without disturbing adsorption sights. However, in complex multi-component systems, such as wastewater, the large presence of alkaline earth ions highly disturbs the adsorption. Even when the solution passes through MZT it continues to pass through the fixed-bed where remaining alkaline earth ions collide with heavy metals, with adsorption sights and among themselves. This all largely disturbs adsorption and influence as well as the concentration gradient which is the driving force for the adsorption.

## 4. Conclusions

Natural zeolite (clinoptilolite) from deposit Vranjska Banja (Serbia) can effectively remove copper from the mono-component (synthetic) solution and the maximum adsorption capacity is almost twice higher when using the flow-through system (~12 mg/g) compared to the batch system (~6 mg/g). The main credits for improved adsorption capacity in the flow-through system are attributed to the constant high concentration gradient which represents the driving force for the adsorption process. Unlike synthetic solutions, wastewater is a multi-component complex system. Associated heavy (Fe, Zn, Mn, etc.) and alkaline earth (Ca and Mg) metals ions, as well as $H^+$ ions, present in large amounts and giving pronounced acidic character to wastewater, are competitive ions for free adsorption sights. The copper removal capacity on zeolite from such a system is negligible and certain pretreatment is obligatory. By increasing the pH value to 4.5 most competitive $H^+$ ions are mostly neutralized and Fe-ions are precipitated while leaving other heavy metals in the solution. After pretreatment, the maximum adsorption capacity in the batch system from wastewater is 4.97 mg/g and this value is up to 80% of the removal capacity from the mono-component system. Adsorption capacity in the saturation point in the flow-through system is 7.05 mg/g, which is 60% of the capacity achieved with the synthetic solution. This means that the presence of accompanying ions more negatively influences flow-through then the batch systems. The easiest way to transfer acquired knowledge about the zeolite and their potential heavy metal removal from the synthetic solutions on the flow-through wastewater system is to perform the batch experiment with synthetic solutions. Based on the results from this study it can be concluded that the approximate value for the breakthrough point in the column fixed-bed wastewater system is around 95% of the maximum adsorption capacity in the batch system.

**Author Contributions:** S.M. (Sonja Milićević) conceived and designed the experiments, wrote the paper, and contributed in analyzing the obtained results; M.V. and S.M. (Sanja Martinović) contributed to the experimental work, especially during flow-through experiments and did the reviews and edits of the paper; M.K. and M.S. performed the characterization of the zeolite; I.J. collected wastewater samples from ZIJIN Bor and characterized them; V.M. contributed during the design of the experiments and in analyzing and explaining of the experimental results. All authors have read and agreed to the published version of the manuscript.

**Funding:** The research presented in this paper was done with the financial support of the Ministry of Education, Science and Technological Development of the Republic of Serbia, within the funding of the scientific research work, according to the contracts with registration numbers 451-03-68/2020-14/200023 and 451-03-68/2020-14/200026.

**Acknowledgments:** We would like to thank to the former management of RTB Bor for providing us with the large number of natural wastewater from the mining deposits.

**Conflicts of Interest:** The authors declare no conflict of interest.

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
