# Peer review of "Removal of Copper from Mining Wastewater Using Natural Raw Material—Comparative Study between the Synthetic and Natural Wastewater Samples"

_minerals, doi:10.3390/min10090753_

Round 1

Reviewer 1 Report

Units and abbreviations of chemical compounds

  • Please unify units for the different fractions of Cli (e.g. lines 20, 106, 179). You should also unify the use of cm3 and dm3 (e.g. lines 146 and 166)
  • Indexes in chemical formulae should be put to subscript (e.g. lines 131-132)

Introduction contains too much general information about heavy metal contamination and adsorption technique. It is not irrelevant to the topic, but all this is known to anyone coming from heavy metal and wastewater research area. Instead you could provide more information about the Cli, and elaborate on differences between synthetic solution and wastewater. The introduction section also lacks a clear aim.

Materials and Methods

  • Paragraph 2.1.1 is a mix of methods and results. I suggest putting all obtained results as a first paragraph under the Results and Discussion section.
  • In the paragraph 2.1.2, which is intended for the characterisation of the wastewater, you also mention the synthetic solution (lines 142-143). I would recommend to modify the title in order to include both types of aqueous media. The Name of the Cu salt given in lines 142-143 is not the same as in line 146.
  • In line 152 you mention that detailed wastewater chemical composition was determined by AAS. Do you mean the chemical composition determined in the supernatant after adsorption experiment with wastewater? Otherwise the sentence as it is now is repetitive to the one given in lines 140-142.
  • The section 2 lacks explanation why only these particular two fractions of Cli were obtained? What happened to the fraction between 0.043 mm and 0.6-0.8 mm? Furthermore, you could mention that all adsorption experiments were carried with the both fractions in paragraph 2.2.
  • Lines 304-305 (KOH-treatment) should be moved to the Materials and Methods, followed by an explanation why it was done.

Results and Discussion

  • Text provided in lines 309-321 could be moved to the Introduction section. It would highlight the problem with Cu regeneration and would justify why your study (utilisation of Cli) is relevant.
  • I would suggest to add a paragraph in Methods and Materials that would include all general information about calculation of isotherms, breakthrough curves, mass zone transfer, etc. The Results and Discussion section now is difficult to read because there is a lot of general information, jumping from the explanations how things were calculated back to the actual results.

Author Response

Thank you for your useful and good review.

Reviewer 2 Report

In this study, copper removal from both synthetic solution and real wastewater using natural zeolite was investigated in batch and continuous flow systems. The data presented in the manuscript are somewhat interesting. The article is suitable for publication in the Minerals Journal when the article will be improved following these comments:

  1. The novelty of this study should be more highlighted in the introduction section.
  2. The maximum adsorption capacity is very low in both batch and continuous flow systems!!! You have to take into consideration these results in large scale application.
  3. Put the results of material characterization in results and discussion section (Not in materials and methods section).
  4. Add XRD and EDS results.
  5. I prefer to use the unit (mg/L) for all chemical concentrations.
  6. Did the experiments conducted in triplicate? Error bars have to be added to all Figures.
  7. For all Figures, the title of axis should be in the middle of that axis.
  8. The experimental conditions should be added to the caption of all Figures.
  9. Add references for all equations used in this paper.
  10. English used in this paper should be carefully polished.
  11. The following important papers should be cited in this article: DOI: 10.1016/j.molliq.2019.111026. DOI: 10.1016/j.molliq.2018.12.115. DOI:10.9734/jsrr/2015/16824

Author Response

(The authors gave the same response as above.)

Round 2

Reviewer 2 Report

I recommend this manuscript in the present form for publication.